# Concept-RidgeAIME: LLM-Guided Automatic Concept-Based Explanations via Ridge-Regularized Inverse Operators for Trustworthy AI

## Abstract

Concept-based explanations replace low-level feature attributions with human-understandable concepts, but existing methods often require model access, heavy computation, or separate procedures for global and local analysis. We extend Approximate Inverse Model Explanations (AIME) and propose Concept RidgeAIME, a model-independent, gradient-free method that provides both global concept importance rankings and local concept contribution vectors via a single ridge-regularized inverse mapping. Our approach represents concepts as shallow rules (e.g., "age $\geq 0.6$," "occupation Exec-managerial $== 1$") that are automatically synthesized by a large language model, then filtered through syntax checks and zero-positive screening for robustness. Concept-RidgeAIME learns inverse operators from model outputs and from concept activations to inputs once, and then computes concept contributions for any instance using simple matrix–vector operations. We evaluated black-box completeness (reconstructability), projection completeness (concept coverage), stability, and latency on the Adult, German Credit, and COMPAS tabular datasets for multiple black-box predictors. The results show that Concept-RidgeAIME delivers accurate, stable, and fast concept-based explanations while unifying global and local views within a single closed-form framework.

## 1 Introduction

With the deployment of high-performance machine learning models in society, the demand to explain *the reasoning behind a decision* using a human conceptual vocabulary has been increasing. Post hoc explanations, represented by methods such as local interpretable model-agnostic explanations (LIME) (Ribeiro et al., 2016) and Shapley additive explanations (SHAP) (Lundberg & Lee, 2017), visualize local feature contributions on the basis of perturbations near the input–output neighborhood. However, because output units remain confined to low-level features such as pixels or one-hot encoding, even experts find it difficult to connect these explanations to a causal or counterfactual understanding of the decision-making process. By contrast, *concept-based explanations* such as TCAV (Kim et al., 2018), ConceptSHAP (Yeh et al., 2020), and concept bottleneck models (CBMs) (Koh et al., 2020) have advantages: they can use human vocabulary (e.g., "highly educated" or "managerial position") to represent feature importance. However, many methods require gradient access or additional training, making it difficult to satisfy both model independence and the simultaneous presentation of global and local information within a single computational framework.

This study reexamines this gap from the perspective of approximate inverse problems. Approximate inverse model explanations (AIME) (Nakanishi, 2023) constructs a single linear operator that approximately inverts the mapping from *output → input* in a least-squares manner, presenting a unified framework for reading its *sequence* (global) and *action* (local). AIME requires no gradient or internal parameters and provides global and local explanations with only one precomputation and matrix–vector multiplication. Consequently, it can be directly applied to gradient-discontinuous tree models and hidden APIs. This study proposes (i) RidgeAIME, which introduces Tikhonov regularization to enhance numerical stability and explanation consistency while preserving these advantages, and (ii) Concept-RidgeAIME, which elevates the explanation

unit from features to human-readable concepts. The latter is novel because it connects the inverse operators of the output-to-input and concept-to-input mappings solely through linear algebra, thereby providing both *global concept formation* and *individual concept contribution (local)* under the same inference rule.

Furthermore, to maintain the design cost of the concepts at a practically acceptable level, this study uses the global feature importance (GFI) from AIME as a scaffold. It then employs a large language model (LLM) to automatically synthesize rule forms for minority literals (e.g., normalization thresholds or one-hot encoding). After generation, the program performs *syntax and data sanitization* by (a) matching feature names, (b) imposing range constraints (e.g., numerical values in $[0, 1]$), and (c) excluding zero-positive rules. Only the *concepts that pass this process* are adopted as the basis for the concept space, thereby minimizing manual trial-and-error while endowing a linear inverse mapping system—which is model-independent, gradient-free, and low-overhead—with concept-level readability.

This study makes three contributions. First, it introduces **RidgeAIME**, which enhances AIME's inverse mapping (Nakanishi, 2023) with Tikhonov regularization to obtain a closed-form inverse that remains numerically well-posed under high correlation, few samples, and many classes. Second, **Concept-RidgeAIME** combines two inverse operators—output-to-input and concept-to-input—to simultaneously provide, through a single linear algebraic readout, (a) global concept rankings and (b) individual concept contribution vectors. Third, it establishes two evaluation metrics: AIME-style *reconstruction-based completeness* (BB $R^2$) and *concept-basis coverage* (projection completeness). Finally, it presents an LLM-assisted concept construction workflow (GFI $\to$ rule generation $\to$ syntactic/zero-positive verification $\to$ adoption) for tabular data. This workflow is intended as a reproducible and importance-anchored procedure for constructing interpretable concepts rather than as a claim of globally optimal concept discovery.

In this sense, Concept-RidgeAIME provides an efficient and model-independent alternative to existing concept-based methods and SHAP-based operational approaches when one seeks a single closed-form framework for global and local concept-level explanations, together with explicit rule-based concept inspection and low inference overhead.

Thus, Concept-RidgeAIME achieves (a) model independence and gradient-free operation; (b) unification of global, local, and conceptual contributions within the same linear framework; (c) completeness (BB projection) and stability (CI); and (d) sub-millisecond execution efficiency (post precomputation). It satisfies these four requirements, providing an efficient and model-independent alternative to existing concept-based methods (e.g., TCAV (Kim et al., 2018), ConceptSHAP (Yeh et al., 2020), and CBMs (Koh et al., 2020)) and SHAP-based methods, while providing explicit and inspectable rule-based explanations with low inference overhead.

Throughout this study, a "concept" refers to a rule-based transformation of input features such as normalized thresholds ($feature \geq t$) or one-hot indicators ($feature == 1$). Thus, each concept corresponds to a shallow decision rule that bundles multiple low-level features into a human-readable unit. This aligns with the widely used notion of rule-based concepts in tabular explainability literature.

In this paper, *feature attribution* refers to local contribution vectors on the original input coordinates, whereas *concept attribution* refers to the re-expression of those local contributions on the adopted rule-based concept basis.

The remainder of this paper is organized as follows: related studies are described in Section 2, the implementation of Concept-RidgeAIME is explained in Section 3, the experiments are discussed in Section 4, and the conclusions are provided in Section 5.

## 2 Related Work

Recent concept-based explanations (concept-based explainable artificial intelligence (XAI)) can be broadly categorized into *(A) post hoc methods measuring the sensitivity and contribution of externally defined concepts*, *(B) methods embedding concepts into the model structure*, *(C) methods using examples or prototypes as concepts*, and *(D) operational approaches applying general feature attribution methods to conceptual representations*. This classification is useful for positioning these approaches relative to each other in terms of

the timing of concept introduction (training/post training), requirements for accessing gradients or intermediate representations, and units of explanation output (global, local, or interventional). A recent systematic survey (Khoozani et al., 2024) traversed this diversifying landscape, organizing key issues around the quality control of concept definitions, evaluation metrics for faithfulness, and connections to automated concept discovery and counterfactual operations.

(A) For post hoc concept sensitivity/contribution, TCAV (Kim et al., 2018) uses directional derivatives with respect to the concept activation vector (CAV) learned from a few positive examples to quantify concept sensitivity for class predictions. Because it requires gradient access and internal representations, it is difficult to apply to nondifferentiable models, such as tree-based models or hidden APIs. However, it has been widely used as a standard method for measuring *global relevance per class*, primarily in the image domain. As an automation of TCAV, Ghorbani et al. (2019) proposed ACE, which first extracts candidate concepts by oversegmenting and clustering images and then assigning importance via TCAV. Methods such as **ICE** (Zhang et al., 2021), which extends CAV from linear to region-based, and **CAR** (Crabbé & van der Schaar, 2022), which generalizes feature regions occupying concepts, are positioned within the trend aimed at improving concept separability and fidelity. The game-theory-based ConceptSHAP defines the sufficiency of concept sets and ensures the axiomatic validity of global importance by allocating marginal contributions via Shapley values; however, it incurs a high computational load owing to combinatorial growth in subset evaluations. Furthermore, **CCE** (Abid et al., 2022), which enables counterfactual debugging at the concept level, constructs *meaningful concept counterfactuals* for each individual to perform causal attributions, thus demonstrating the feasibility of ex post concept manipulation.

(B) For concept internalization (during learning), **CBMs** (Koh et al., 2020) explicitly predict concepts at intermediate layers and infer final labels on top of them, thereby reconciling *interventional capability* (rewriting concepts to control output) and *global and local explanations. Post hoc CBM* has also been proposed for pretrained black-box models (Yüksekgönül et al., 2022). **SENN** (Alvarez-Melis & Jaakkola, 2018) simultaneously learns "interpretable base concepts" and linear readouts, ensuring separability and stability through regularization. **Concept whitening** (Chen et al., 2020) enhances the interpretability of internal representations by inserting a whitening layer into convolutional neural networks (CNNs) to align latent axes with known concepts. Although internalization methods require concept supervision or retraining, they offer the advantage of providing highly coherent explanations through *intervention/constrained learning.*

(C) As a method for treating examples or prototypes as concepts, **ProtoPNet** (Chen et al., 2019) performs classification based on similarity to *prototypes (typical patches)* learned per class, presenting *example-based local explanations* of the "this looks like that" type. **Net2Vec** (Fong & Vedaldi, 2018) quantifies the correspondence between filters and concepts, whereas **Concept Attribution** (Wu et al., 2020) constructs *global explanations* for CNNs. Furthermore, Kumar et al. (2021) and Kamakshi et al. (2021) proposed implementation approaches—MACE and PACE, respectively—that extract and summarize concepts from visual models as model- and architecture-independent *posterior concept extractors.*

(D) Regarding the application of generic attribution methods to concept representations, **Concept Space SHAP** is not a formally defined method but rather an operational approach that treats *existing concept representations* (such as rule scores or one-hot concepts) as features and applies generic SHAP (Lundberg & Lee, 2017). Although it yields *local and global contributions* based on the Shapley axioms, the explanation quality depends on the *computational load of the sampling approximation* and the *fidelity and granularity of concept representations.* In contrast, applying SHAP (Lundberg & Lee, 2017) to *arbitrary concept representations* is also common; however, this is merely an *operational practice* of replacing SHAP features with concept features (user-designed or automatically extracted) and is not a distinct method name. Similarly, whereas local Shapley values per concept can be obtained, computational overhead from sampling or approximation remains unavoidable. Moreover, it does not provide *global, local, or inter-concept mappings* within a unified linear framework.

The *Concept-RidgeAIME* method proposed in this study extends the approximate inverse operator of AIME (Nakanishi, 2023) to the concept space. It simultaneously presents *global (concept formation) and local (concept contribution) explanations through a single linear algebra operation* in a *closed form with regularization* while remaining *model-independent and gradient-free.* Postprocessing systems (TCAV, ACE,

ConceptSHAP, CCE) provide concept relevance, completeness, and counterfactuality but have computational and access constraints that limit their use. Internalization systems (CBM, SENN, concept whitening) offer intervenability and high fidelity but require retraining. Concept Space SHAP is versatile but depends on the computational cost of approximation and concept representation quality. By contrast, Concept-RidgeAIME enables the evaluation of *BB completeness (reconstruction $R^2$)* and *concept projection completeness (Projection)* within a single implementation via *one-time precomputation and matrix–vector multiplication* and adapts to *gradient-independent operational environments* such as tabular, tree-based, and confidential APIs. Furthermore, to address the challenges identified in the survey (Khoozani et al., 2024) (quality control and automatic discovery of concept definitions, connection with counterfactual explanations), this study combines *lightweight rule synthesis and validation using LLMs* with *reconstruction evaluation of linear inverse mappings*, making the *concept design, verification, and presentation* reproducible, which holds practical significance.

Yu et al. (2025) comprehensively reviewed the challenges and solutions for achieving "trustworthiness" in LLM-based agents and multiagent systems, systematizing all aspects from attack and defense methods to evaluation approaches under the TrustAgent framework. Mumuni & Mumuni (2025) comprehensively organized the development of XAI, considering structurally explainable models, black-box models, and even automated explanation generation by using LLMs and vision-language models, presenting their strengths and challenges. Bilal et al. (2025) comprehensively discussed the latest technologies, applications, and evaluation methods for utilizing LLMs in XAI, presenting user-friendly explanation generation, evaluation, and prospects for real-world applications. Furthermore, Basheer et al. (2025) proposed a framework for predicting vulnerability patches via LLM-based BERT models in the cybersecurity domain and discussed practical applications integrated with reliability enhancements. Benk et al. (2025) investigated user expectations and values regarding LLM reliability standards, identified a perception gap between developers and users, and examined the challenges for the standardization of these models. These studies demonstrate the rapid expansion of XAI initiatives incorporating LLMs. However, most remain at the conceptual organization or application stage, and challenges persist in establishing an implementation foundation that uniformly provides global-, local-, and conceptual-level explanations within a model-independent framework.

By contrast, our approach is unique in that Concept-RidgeAIME, an extension of AIME, simultaneously achieves explanations across these three layers via only linear algebra operations, which are independent of gradients or internal representations. Furthermore, by combining LLM-based automatic rule generation with consistency verification, a reproducible framework that streamlines the design of the conceptual space is provided, thereby addressing the limitations of existing research.

Overall, Concept-RidgeAIME simultaneously achieves (a) model and gradient independence; (b) simultaneous presentation of global, local, and conceptual contributions; (c) completeness evaluation based on reconstruction metrics; (d) numerical stability and reproducibility through regularization; and (e) operational feasibility of concept design via LLM assistance. Thus, it complements the strengths of TCAV, ConceptSHAP, CBM, and Concept Space SHAP while filling practical gaps in black-box explanations for tabular data (gradient independence, low computational load, and local concept contribution).

## 3 Method

In this study, concepts are defined as rule-based transformations of the input, either normalized thresholds (feature $\geq t$) or one-hot indicators (feature $== 1$). These align with the standard notion of rule concepts widely used in tabular XAI. In this work, a concept is treated as an explicit and inspectable rule-based unit rather than as a latent semantic construct.

An overview of the proposed framework is illustrated in Appendix Figure 4, which summarizes two-stage inverse mapping (RidgeAIME and Concept-RidgeAIME) together with the LLM-assisted concept-construction pipeline.

**Notation.** Throughout this paper, pinv($\cdot$) denotes the Moore–Penrose pseudoinverse. We use $M_{i:}$ and $M_{:j}$ to denote the $i$-th row and $j$-th column of a matrix $M$, respectively. For a matrix $M$, $M_{i:}$ and $M_{:j}$ denote the $i$-th row and $j$-th column, respectively. The symbol $\ell_i$ denotes the local feature contribution vector for

sample $i$, and $v_i$ the corresponding concept contribution obtained from $U_\gamma^\top \ell_i$. The operator $\Pi_C$ denotes the orthogonal projection onto the column space of $U_\gamma$.

This section extends the framework of AIME (Nakanishi, 2023) to a notation where samples are arranged in rows. $X \in \mathbb{R}^{d \times n}$ ($d$-dimensional features, $n$ samples), $Z \in \mathbb{R}^{k \times n}$ ($k$-dimensional downstream representation, including class one-hot, logit, intermediate activations). The core function of AIME is to approximate the unknown forward linearization $A \in \mathbb{R}^{k \times d}$ under the assumption that it yields $z \approx Ax$. This approximation is performed *from the output to the input side* by estimating the approximate inverse operator $A^\dagger \in \mathbb{R}^{d \times k}$ that maps from data $(X, Z)$ to $z \approx Ax$ via least-squares estimation. This single linear operator simultaneously extracts global and local explanations.

In column-vector notation, the approximate inverse operator $W \in \mathbb{R}^{d \times k}$ is defined as

$$\min_{W \in \mathbb{R}^{d \times k}} \ \big\| X - WZ \big\|_F^2 \quad \Longrightarrow \quad W^\star = X Z^\dagger = A^\dagger, \tag{1}$$

where $\| \cdot \|_F$ denotes the Frobenius norm. This formulation corresponds to the minimum-norm solution using the Moore–Penrose pseudoinverse $Z^\dagger$. Here, $A$ denotes the conceptual forward operator representing the local linearization (mean Jacobian) of the model, and is directly approximated from the data matrix as $A^\dagger \approx X Z^\dagger$.

**Global/Local Readout.** Once the approximate inverse operator $W^\star = A^\dagger$ is obtained, the GFI (a $k$-dimensional weight vector for feature $j$) can be derived from $(W^\star)_{j:} \in \mathbb{R}^{1 \times k}$ (row $j$), with the scalar importance metric $\|(W^\star)_{j:}\|_p$ (e.g., $p = 2$). The local feature contribution of sample $i$ (column vector $x_i \in \mathbb{R}^d$, $z_i \in \mathbb{R}^k$) is

$$\ell_i := \big(W^\star z_i\big) \circ x_i \in \mathbb{R}^d, \tag{2}$$

where ($\circ$ denotes the Hadamard product). The sign indicates the direction of change (boost or suppression).

### 3.1 RidgeAIME: Approximate Inverse with Tikhonov Regularization

The pseudoinverse $Z^\dagger$ may become numerically unstable under conditions such as high correlation, few samples, or many classes. Therefore, Tikhonov (ridge) regularization is applied as follows:

$$\min_W \big\| X - WZ \big\|_F^2 + \lambda \|W\|_F^2, \qquad \lambda \geq 0, \tag{3}$$

with the regularized solution

$$W_\lambda = X Z^\top (ZZ^\top + \lambda I_k)^{-1} = X Z_\lambda^\dagger = A_\lambda^\dagger, \tag{4}$$

where $Z_\lambda^\dagger := Z^\top (ZZ^\top + \lambda I_k)^{-1}$ denotes the Tikhonov pseudoinverse. As $\lambda \to 0$, this reduces to $W_\lambda \to W^\star$.

Equation (4) follows directly from the normal equations of ridge regression applied to the Frobenius-norm least-squares formulation in (3). Specifically, minimizing $\|X - WZ\|_F^2 + \lambda \|W\|_F^2$ yields $W ZZ^\top + \lambda W = X Z^\top$, whose closed-form solution is $W_\lambda = X Z^\top (ZZ^\top + \lambda I_k)^{-1}$. This is a standard result of Tikhonov-regularized linear least-squares , which we now cite explicitly.

The closed-form expression in (4) requires that $(ZZ^\top + \lambda I_k)$ is symmetric positive definite, which holds for any $\lambda > 0$. Therefore, no invertibility assumption on $ZZ^\top$ itself is needed. When $\lambda = 0$, the solution reduces to the Moore–Penrose pseudoinverse via SVD, which remains well-defined even when $ZZ^\top$ is singular.

The readout rule is identical to equation 2, with $W^\star$ replaced by $W_\lambda$:

$$\text{GFI (feature } j\text{): } (W_\lambda)_{j:}, \qquad \text{LFI (sample } i\text{): } \ell_i := (W_\lambda z_i) \circ x_i. \tag{5}$$

This regularization improves the condition number of $W_\lambda$, enhancing robustness against outliers and noise and increasing the reproducibility of the GFI rankings.

Alternative regularizers such as L1 or elastic-net could enforce sparsity, but they typically require iterative optimization and make the signs and relative magnitudes of coefficients sensitive to hyperparameter tuning.

We found the direct interpretation of coefficients as feature importance—a characteristic of this explanation method—to be undesirable. Ridge regularization, by contrast, preserves a closed-form solution, keeps the geometry of contribution vectors smooth, and empirically yields the most stable rankings across bootstrap samples.

### 3.2 Concept-RidgeAIME: Two-Stage Extension to Concept Space

To use *human-readable concepts* rather than features as the units of explanation, A two-stage approximate inverse is constructed while retaining the columnwise notation.

**Concept Score Matrix.** The concept score matrix $C \in \mathbb{R}^{q \times n}$ ($q$ concepts) is evaluated for each sample from LLM-generated or manually defined rules. Numerical features are handled using normalization thresholds, and categorical features are represented via one-hot encoding.

**Approximate Inverse from Concept to Feature.** To re-express the local feature contribution vector $\ell_i$ on this concept basis,

$$\min_{U \in \mathbb{R}^{d \times q}} \left\| X - UC \right\|_F^2 + \gamma \|U\|_F^2 \quad \Longrightarrow \quad U_\gamma = X C^\top (CC^\top + \gamma I_q)^{-1}. \tag{6}$$

Here, $(U_\gamma)$ is the approximate inverse operator mapping "concepts to $t$ features."

**Local Concept Contribution.** Mapping $\ell_i$ from equation 5 to the concept basis yields

$$v_i := U_\gamma^\top \ell_i \in \mathbb{R}^q, \qquad \text{ratio}_{i,c} := \frac{v_{i,c}}{\sum_{c'=1}^q |v_{i,c'}|}. \tag{7}$$

Here, $v_{i,c} > 0$ ($< 0$) indicates that concept $c$ boosts (suppresses) the judgment, and $i$ indexes the sample, with $i = 1, \ldots, n$. The ratio ratio represents the relative contribution of each concept and is highly interpretable.

### 3.3 Completeness Metric

The adequacy of the explanation is evaluated using two $R^2$ metrics on the basis of the AIME reconstruction accuracy. Specifically, the Frobenius norm is used in BB completeness to capture the overall reconstruction error across all samples and features: where the Euclidean norm (2-norm) is used in projection completeness to measure the fidelity of each individual contribution vector when projected onto the concept basis.

BB completeness (reconstruction $R^2$) measures how well the inverse operator reconstructs the inputs from the model outputs, whereas projection completeness evaluates how much of each local explanation vector $\ell_i$ can be expressed within the conceptual subspace spanned by $U_\gamma$.

Specifically, the Frobenius norm is used in BB completeness to capture the overall reconstruction error across all samples and features, whereas the Euclidean norm ($\ell_2$-norm) is used in projection completeness to measure the fidelity of each individual contribution vector when projected onto the concept basis.

**BB Completeness (Output→Input).**

$$R_{\text{BB}}^2 := 1 - \frac{\left\| X - W_\lambda Z \right\|_F^2}{\left\| X - \bar{X} \right\|_F^2}, \qquad \bar{X} = \frac{1}{n} X \mathbf{1}\mathbf{1}^\top, \tag{8}$$

where $\mathbf{1}$ denotes a vector of all 1 s with length $n$). For $\lambda = 0$, this coincides with AIME in $R^2$.

**Projective Completeness (Feature Contribution to Conceptual Basis).**

$$R_{\text{Proj}}^2 := 1 - \frac{\sum_{i=1}^n \left\| \ell_i - U_\gamma v_i \right\|_2^2}{\sum_{i=1}^n \|\ell_i\|_2^2} = 1 - \frac{\sum_{i=1}^n \left\| \ell_i - \Pi_{\mathcal{C}}(\ell_i) \right\|_2^2}{\sum_{i=1}^n \|\ell_i\|_2^2}, \tag{9}$$

($\Pi_{\mathcal{C}}$ is the orthogonal projection of $\text{span}(U_\gamma)$); because $U_\gamma$ is obtained via least squares, $U_\gamma v_i$ typically coincides with the optimal projection.

### 3.4 Numerical Implementation and Computational Complexity Key Points

The operator $W_\lambda = A_\lambda^\dagger = XZ^\top(ZZ^\top + \lambda I_k)^{-1}$ is obtained by solving a symmetric positive-definite system of size $k \times k$, which can be computed quickly and stably. The local description requires only matrix–vector multiplication $W_\lambda z_i$ and elementwise multiplication per column, taking $\mathcal{O}(dk)$. Conceptually, $U_\gamma$ is a $q \times q$ system, which is particularly lightweight when $q \ll d$. In terms of implementation, (i) column centering and normalization of $Z$; (ii) scale normalization of $\lambda$ (e.g., $\lambda = \alpha, \mathrm{tr}(ZZ^t op)/k)$; and (iii) SPD solvers such as Cholesky (falling back to pinv upon failure) are effective. Because it does not rely on the model's internal gradients or parameters, the method is *completely model independent.*

### 3.5 Setting the Regularization Coefficient $\lambda$ in RidgeAIME

**Background and Role.** AIME/RidgeAIME learns a linear operator that performs an "approximate inverse mapping" from the model outputs (or downstream representations) to the inputs. The notation herein represents each sample as a column vector, such that $X \in \mathbb{R}^{d \times n}$ denotes the inputs and $Z \in \mathbb{R}^{k \times n}$ denotes the downstream representations, where $n$ is the number of samples. The approximate inverse operator of "$Z \to X$" is defined as

$$A_\lambda \in \mathbb{R}^{d \times k}$$

. RidgeAIME solves the following *Tikhonov-regularized* least-squares problem:

$$\min_{A \in \mathbb{R}^{d \times k}} \left( \|X - AZ\|_F^2 + \lambda \|A\|_F^2 \right), \qquad \lambda \geq 0. \tag{10}$$

with the closed-form solution

$$A_\lambda = XZ^\top(ZZ^\top + \lambda I_k)^{-1} = X Z_\lambda^\dagger, \qquad Z_\lambda^\dagger := Z^\top(ZZ^\top + \lambda I_k)^{-1}, \tag{11}$$

where $I_k$ is the $k \times k$ identity matrix. As $\lambda \to 0$, $A_\lambda \to XZ^\top(ZZ^\top)^{-1}$ (when $Z$ is full rank), matching the unregularized AIME. The readout for RidgeAIME is identical to that in the previous section. GFI corresponds to the column vectors of $A_\lambda$, and the local feature importance (LFI) for sample $i$ is

$$\ell_i = (A_\lambda z_i) \circ x_i \in \mathbb{R}^d \tag{12}$$

where $z_i$ and $x_i$ denote the $i$th columns of $Z$ and $X$, respectively, and $\circ$ denotes the elementwise product. Regularization improves the condition number of $(ZZ^\top)$, enhances numerical stability, and mitigates the effects of outliers and multicollinearity.

**Policy for $\lambda$ in this implementation (fixed value).** In the submitted program, *reproducibility and computational efficiency were prioritized.* For all datasets (Adult, German Credit, COMPAS),

$$\lambda = 10^{-3}$$

The same order of magnitude is used even when combined with the conceptual linear mapping setting. All features are normalized to the scale $[0, 1]$ (min–max for continuous variables and one-hot for categorical features). Under this scale, $\lambda = 10^{-3}$ (i) keeps the condition number of $ZZ^\top + \lambda I_k$ within a safe range, (ii) does not degrade BB completeness ($R_{\mathrm{BB}}^2$) or projective completeness ($R_{\mathrm{Proj}}^2$), and (iii) Suppresses the bootstrap-induced variance. This practical trade-off confirms that it is a stable choice. *Preliminary verification.* A grid search was not performed; a fixed value was used throughout.

**Notes on Numerical Implementation.** equation 11 requires solving only a single $k \times k$ symmetric positive-definite system, which can be computed quickly and stably via Cholesky decomposition (falling back to pseudoinverse via SVD if it fails). Furthermore, column centering and scaling of $Z$, together with alignment of the one-hot basis, improve the estimation of $A_\lambda$ and the stability of $\ell_i$.

## 4 Experiments

The effectiveness of Concept Ridge AIME was evaluated on tabular datasets (Adult/German Credit/COMPAS) using four criteria: (i) black-box reproducibility completeness (**BB Completeness**: $R^2$ when externally fitting the black-box logit model using only the concept matrix $C$); (ii) projection completeness of local contributions onto the conceptual space (**Projection Completeness**: norm ratio $\|\Pi_{\mathcal{C}}\ell\|_2/\|\ell\|_2$ when projecting the AIME-derived local feature contribution vector $\ell$ onto the concept basis); (iii) stability of the metric (**95% CI**: estimated from 200 bootstrap samples); and (iv) computational efficiency (**Latency**: average inference time per instance, including warm-up).

**Why BB/Projection metrics are reported only for Concept-RidgeAIME.**

BB completeness and projection completeness measure the quality of linear reconstruction and projection onto a linear conceptual basis. These metrics are defined only for inverse operator methods and are not applicable to ConceptSHAP, TCAV, CBM, or Kernel/Tree SHAP because these methods do not provide linear reconstruction of inputs or concept-space projections. Therefore, they are marked as "–" in Table 1.

**Experimental Setup**

For each dataset, missing values were imputed using the median for numeric features and the mode for categorical features. Numeric values were normalized to $[0,1]$, and categorical features were one-hot encoded. Training and evaluation followed a fixed stratified 8:2 split. The black-box model was LightGBM (learning rate = 0.05, number of trees = 300, other parameters set to defaults). Concepts were generated by first extracting the top features and one-hot indicators from AIME's GFI and then feeding them into an LLM context to generate AND/OR rules (thresholds in $[0,1]$; one-hot indicators expressed as ==1). Rules were adopted after syntactic validation (feature-name matching and exclusion of zero or positive cases). If fewer than six rules were generated, backup concepts based on quantile points were automatically added to ensure concept set coverage. In COMPAS, race and gender were excluded from the predictive concept set and treated separately as auxiliary concepts for auditing. The comparison methods include Concept-SHAP (concept Shapley), CBM (two-stage logistic), TCAV (finite difference approximation), Concept Space SHAP (Ridge approximation + KernelExplainer), Feature SHAP (Tree/Kernel), and LIME. All of these models can be deployed and managed via a common model-agnostic API. Finally, the inverse operator for AIME/ConceptAIME was computed only once, and subsequent outputs were obtained solely through matrix–vector operations, yielding inference speeds several orders of magnitude faster.

This concept-construction stage is intended to produce a transparent and importance-anchored basis rather than a globally optimal rule set. In other words, the role of the LLM/GFI pipeline in this paper is not to maximize predictive performance over arbitrary candidate rules but to construct a reproducible and semantically inspectable concept basis whose faithfulness can then be quantified through BB completeness and projection completeness. For this reason, we do not position the present workflow as evidence of globally optimal concept discovery relative to random rule sets.

**Results and Discussion**

Table 1 reports *BB completeness ($R^2$) + projection completeness + 95% CI + inference time (ms/instance)* across the three datasets. For adults, BB completeness was high at 0.725222 (95% CI: 0.723265–**0.727076**), indicating that even with only six concepts, the black-box logit could be well-reproduced externally. Projection completeness was also high at **0.851456** (**0.834869**–**0.865143**), indicating that the local contribution $\ell$ adheres closely to the conceptual basis with high precision and minimal information loss. German Credit exhibited a BB completeness of **0.339209** (**0.324459**–**0.352034**), whereas projection completeness was stably high at **0.828336** (**0.822406**–**0.834705**). Therefore, even with only hard binary rules, local explanations can be sufficiently expressed in the conceptual space. However, the external reproducibility of black-box logits could be improved by expanding the number of concepts or using softened (probabilistic) outputs. COMPAS showed a BB completeness of **0.221209** (**0.216630**–**0.225862**), but projection completeness was the highest at **0.901333** (**0.893052**–**0.908658**), demonstrating that local contributions can

be compressed extremely efficiently into conceptual coordinates. The inference time for ConceptAIME remained approximately **0.013–0.014 ms/instance** for all datasets, which is significantly faster than that of ConceptSHAP (**60.6–109.7 ms**), Concept Space SHAP (**54.9–133.9 ms**), LIME (**177.5–323.7 ms**), and TCAV (**2.55–2.64 s** on Adult/German Credit). CBM was also fast (1–1.6 ms), but ConceptAIME is fundamentally different in that it can *present both local and global information simultaneously without modifying the trained black box*. Figs 1, 2, and 3 present the global rankings of rule-based concepts obtained via ConceptAIME, ConceptSHAP, CBM, TCAV, Concept Space SHAP, and Feature SHAP for the Adult, German Credit, and COMPAS datasets. In contrast to high-level human labels, the updated figures display each concept in its explicit rule-based form (e.g., "$marital - status\_Married - civ - spouse == 1$", "$age \geq 0.60$", "$checking\_status\_ < 0 == 1$"). This makes the differences among concept definitions transparent and enables direct alignment between each rule and its quantitative impact. Across datasets, ConceptAIME consistently identifies interpretable and dataset-specific rules, such as age thresholds, one-hot indicators of marital or employment status, and credit status flags, demonstrating that linear inverse mappings can extract domain-relevant concept structures without manually engineered concept vocabularies.

As shown in Appendix Table 9, we conducted a ridge ablation across $\lambda \in \{0, 10^{-6}, 10^{-3}, 10^{-1}\}$ on the Adult and German datasets. Both BB completeness and projection completeness remained effectively invariant across all tested values, suggesting that the proposed inverse formulation is robust to the choice of regularization strength in the tested range and reduces the need for delicate hyperparameter tuning in practice.

Table 1: BB completeness ($R^2_{\mathrm{BB}}$), projection completeness ($R^2_{\mathrm{Proj}}$), and latency for Concept-RidgeAIME and baselines.

| Dataset | Metric | Mean | 95% CI |
|---|---|---|---|
| Adult | BB completeness | 0.725222 | [0.723265, 0.727076] |
| | Projection completeness | 0.851456 | [0.834869, 0.865143] |
| | Latency (ConceptAIME) | 0.013388 | – |
| German Credit | BB completeness | 0.339209 | [0.324459, 0.352034] |
| | Projection completeness | 0.828336 | [0.822406, 0.834705] |
| | Latency (ConceptAIME) | 0.013841 | – |
| COMPAS | BB completeness | 0.221209 | [0.216630, 0.225862] |
| | Projection completeness | 0.901333 | [0.893052, 0.908658] |
| | Latency (ConceptAIME) | 0.013638 | – |
| **Baseline Latencies (ms/instance)** | | | |
| **ConceptSHAP** | Adult/German/COMPAS | 109.71/108.19/60.64 | |
| **CBM** | Adult/German/COMPAS | 1.07/1.25/1.58 | |
| **TCAV** | Adult/German/COMPAS | 2550.51/2635.22/– | |
| **C-space SHAP** | Adult/German/COMPAS | 133.94/98.58/54.90 | |
| **SHAP** | Adult/German/COMPAS | 5.70/4.52/3.37 | |
| **LIME** | Adult/German/COMPAS | 323.67/177.51/187.92 | |

Overall, the experimental results support three concrete observations: (1) high projection completeness, indicating that the local feature contribution vectors can be expressed with little loss in the adopted concept basis; (2) nontrivial BB completeness on all three datasets, showing that a small set of explicit rules can recover a meaningful fraction of the black-box output behavior; and (3) sub-millisecond inference after one-time precomputation, which is substantially faster than the sampling-based baselines considered here. These results support the method as a model-independent and computationally light framework for concept-level explanation. For datasets such as German Credit and COMPAS, where strong nonlinear interactions are suspected, the high projection metric indicates that conceptual coordinate explanations remain informative even when exact black-box reconstruction is more challenging. We do not claim that the present LLM/GFI pipeline discovers an optimal concept basis; rather, it provides a transparent and reproducible basis whose faithfulness is quantified through BB completeness and projection completeness.

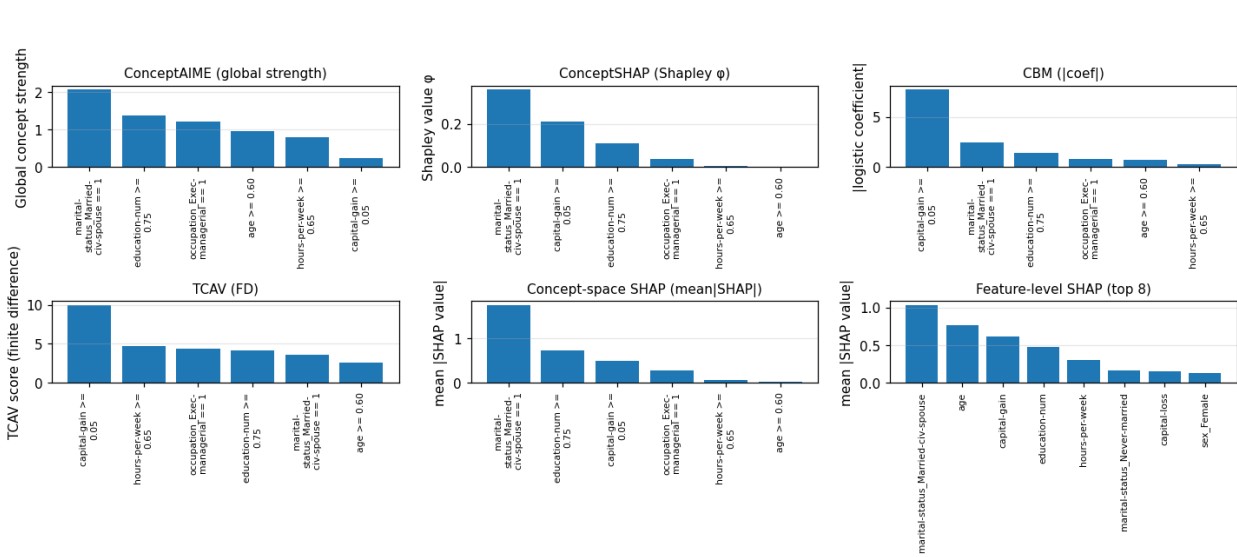

Figure 1: Global concept rankings (Adult). Each bar corresponds to an explicitly defined rule-based concept (e.g., "marital-status_Married-civ-spouse == 1", "age ≥ 0.60"). The vertical axis represents the aggregated global concept strength, $\|U_\gamma^\top \ell\|_1$.

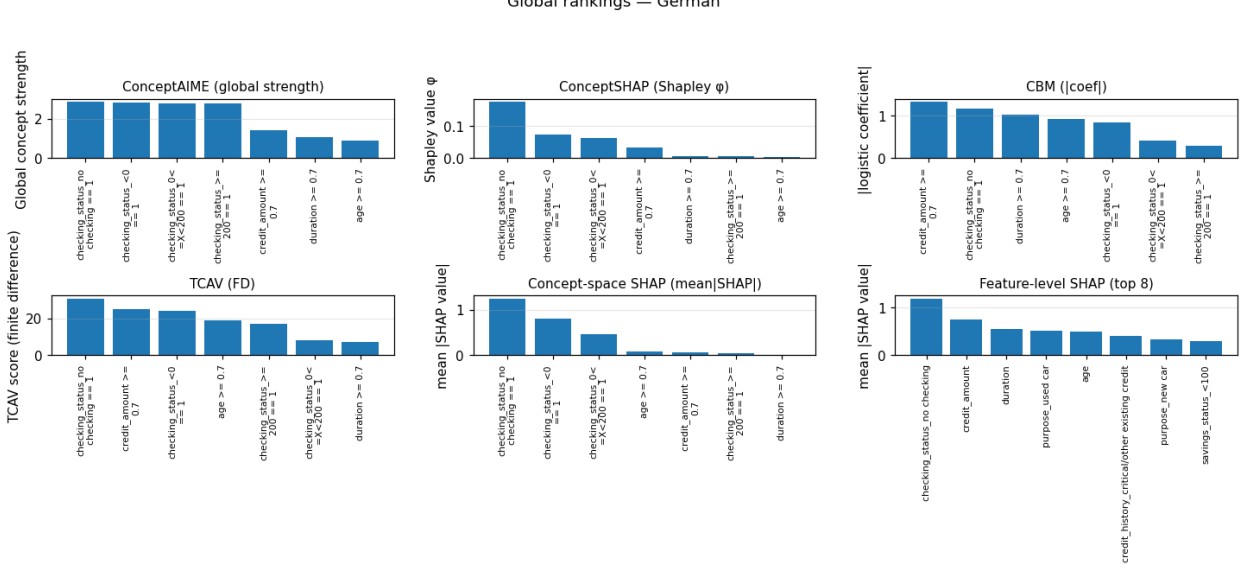

Figure 2: Global concept rankings (German Credit). Each bar shows the impact of rule-based concepts such as "*checking_status_nochecking == 1*" or "*credit_amount ≥ 0.7*". The vertical axis represents the aggregated global concept strength, $\|U_\gamma^\top \ell\|_1$.

Although the rule-based concepts resemble depth-1 decision stumps, Concept-RidgeAIME does not reimplement the black-box model. Instead, the linear inverse operator maps from logits to input features, and the rule concepts form an independent conceptual basis. Therefore, similarity in structure does not imply

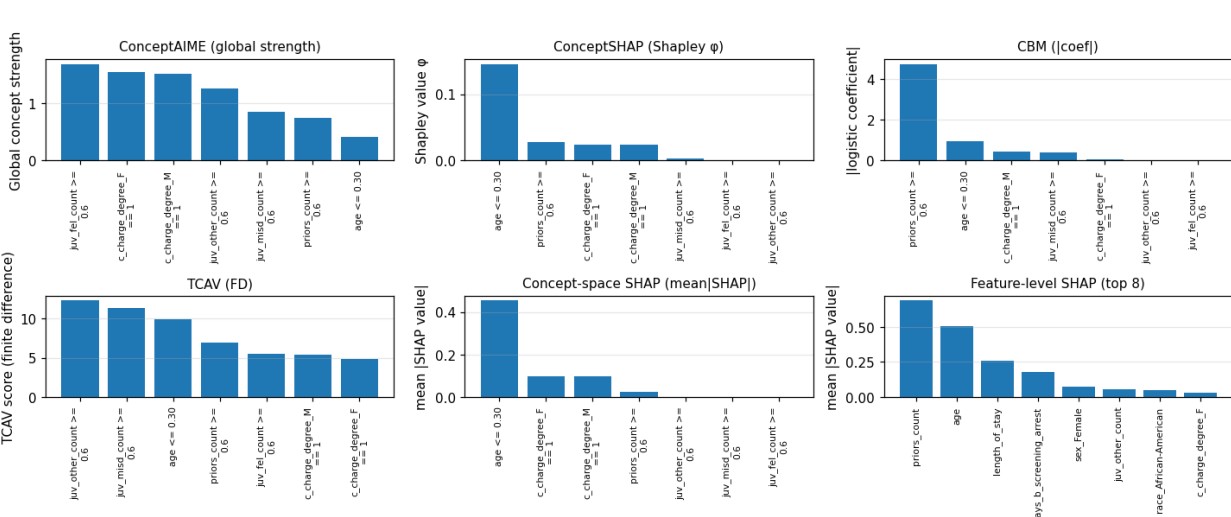

Figure 3: Global concept rankings (COMPAS). Concepts correspond to explicit rules (e.g., "$age \leq 0.30$", "$juv\_fel\_count \geq 0.6$", "c_charge_degree_F == 1"). The vertical axis represents the global concept strength, $\|U_\gamma^\top \ell\|_1$.

information leakage from the black box: the projection completeness metric confirms that local contributions remain faithful even when the black-box model is a gradient-discontinuous tree ensemble.

The present experiments focus on tree-based black-box models (LightGBM). The method is fully model independent and has been applied to neural models in prior AIME work. Because Concept-RidgeAIME requires only downstream representations $Z$, the same pipeline applies to MLPs, logistic models, and transformer-based tabular models without modification.

## 5   Conclusion

This study proposes Concept-RidgeAIME, which inherits the inverse problem perspective of AIME (Nakanishi, 2023), enhances numerical stability and reproducibility through Tikhonov (ridge) regularization, and further extends the framework to the conceptual space. The method comprises two stages: an approximate inverse operator $W_\lambda$ for output-to-input mapping and an approximate inverse operator $U_\gamma$ for concept-to-input mapping. This design uniquely enables the consistent extraction of *global*, *local*, and *concept-level* contributions within a single linear algebra representation. Although preserving the advantages of AIME (model independence, no gradient requirement, and millisecond-level inference after batch precomputation), it additionally enables decision-making to be described via higher-level conceptual units designed by a user or an LLM, thereby achieving interpretable explanations aligned with expert vocabulary.

Comparatively, ConceptSHAP excels in *global importance* based on game-theoretic axioms but faces challenges in terms of computational load from subset evaluation and the design of local explanations. TCAV presents sensitivity-based class-specific concept relevance but requires access to internal representations and gradients. CBM provides *interventional* internalized explanations but requires concept supervision during training, making retrofitting onto existing black-box models difficult. Concept Space SHAP is an *operational* approach that applies SHAP to concept representations, inheriting the underlying theory but retaining the computational burden of sampling approximations. By contrast, Concept-RidgeAIME achieves *(i) model and gradient independence*; *(ii) simultaneous presentation of global, local, and conceptual contributions*; *(iii) quantification of fidelity using two completeness metrics (BB and projection)*; and *(iv) numerical stability*

*and reproducibility via regularization.* This fills practical gaps in black-box explanations for tabular data, combining low computational load with local conceptual contributions.

Furthermore, this study prioritizes the operational feasibility of concept design by automatically synthesizing concept candidates with LLM assistance to reduce manual effort. By strictly constraining the rule format (only known feature names, thresholds in $[0, 1]$, one-hot equality, JSON-only) and performing syntax checks and zero/positive exclusion at the program level, this study enables a workflow of *automatic generation* → *verification* → *adoption* to be reproduced within a single notebook, enhancing both the reproducibility and portability of the results.

However, this study has several limitations. First, *dependency on concept set quality*: inappropriate thresholds or extremely sparse rules reduce BB and projection completeness and introduce bias into local explanations. Second, *linear approximation limitations*: because the method relies on linear approximation, strong nonlinear interactions may be under- or overestimated. Third, *scaling assumptions*: dependence on $[0, 1]$ normalization and one-hot encoding, which means that stability can be compromised by improper preprocessing. Fourth, *LLM generation validation*: verification and review are essential to prevent the introduction of unknown features or complex logic (e.g., OR or negation). A further limitation is that the semantic validity of the generated concepts, in the sense of alignment with domain experts' conceptual understanding, has not been directly evaluated in this work. Instead, concept quality is assessed through explicitness, reproducibility, and faithfulness, quantified by BB completeness and projection completeness. In addition, the experiments are currently limited to tabular datasets with tree-based black-box models, although the framework itself is model-independent and may be extended to neural and transformer-based settings in future work.

## Reproducibility Statement

To ensure reproducibility, a comprehensive description of the proposed method is provided, including mathematical formulations (Sections 2–4), hyperparameter settings (Section 4), and implementation notes (Appendix A). All datasets used (Adult, German Credit, and COMPAS) are publicly available. All the implementation details and source code, including a single executable notebook `Auto_Concept_AIME_ICLR2026.ipynb` that automatically downloads the datasets, trains the models, performs concept generation, and reproduces all the figures and tables, are provided as anonymized supplementary material. Bootstrap confidence intervals are also reported for all key metrics to quantify robustness. These measures collectively ensure that independent researchers can reproduce both the methodology and results presented in this paper in a fully self-contained manner.

## Broader Impact Statement

This work aims to enhance the transparency, fairness, and trustworthiness of machine learning systems. by providing human-understandable, concept-based explanations at both the global and local levels. Because Concept-RidgeAIME operates without accessing model internals or gradients, It can be applied to a wide range of real-world black-box systems in domains such as healthcare, finance, and public administration. where interpretability and accountability are essential.

Potential negative impacts include the risk of misunderstanding or overtrusting explanations that appear intuitive but may not fully reflect model behavior. and the possibility of biased or misleading concept definitions when LLMs are used for concept generation. To mitigate these risks, the method enforces strict syntactic validation, excludes trivial zero/positive rules, and supports deterministic, auditable concept generation. This study does not automate decision-making itself but instead provides tools to support human auditing and understanding of AI systems.

Promoting model transparency and concept-level interpretability while maintaining awareness of potential biases and misuse can contribute to the responsible and ethical deployment of AI technologies.

**Author Contributions**

The sole author undertook the conception of the study, the development of the methodology, the implementation of the algorithms, the design and analysis of the experiments, and the complete writing and revision of the manuscript.

**Acknowledgments**

The author thanks Editage [www.editage.com] for English language editing.

**Use of Generative AI**

The proposed method incorporates an LLM, utilizing the GPT4-o mini as its API. Additionally, GPT-5 Pro, DeepL, and Paperpal were employed to assist with implementing the method and refining the text of this paper.

The author takes full responsibility for the content of this paper.

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

## A Overview of Concept-RidgeAIME

For clarity, we provide a schematic overview of the proposed framework, summarizing the overall pipeline and the interaction between inverse mapping and concept construction (Fig. 4).

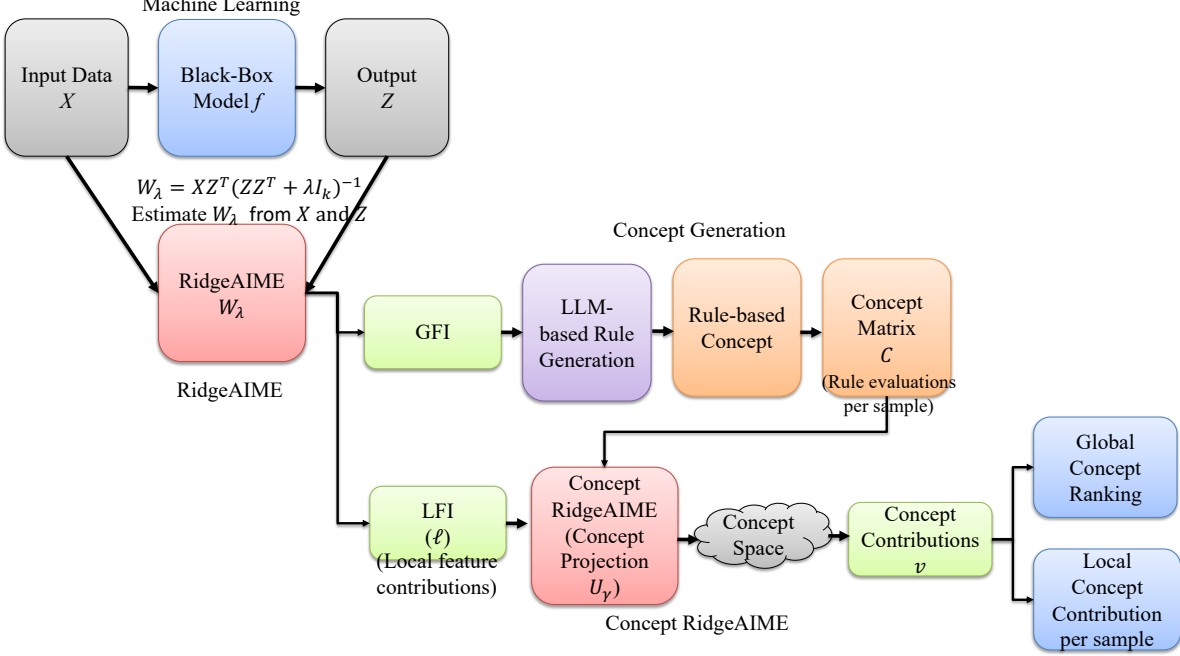

Figure 4: Overview of the proposed Concept-RidgeAIME framework. RidgeAIME constructs an approximate inverse mapping from model outputs to inputs, producing global feature importance and local feature contributions. Concept-RidgeAIME then projects these contributions onto a rule-based concept space constructed via an LLM-assisted pipeline, enabling global and local concept-level explanations.

## B Reproduction Steps and Environment.

Implemented and tested on Google Colab Pro+ (Python 3.12.11). The experiments used LightGBM 4.5.0, scikit-learn 1.5.2, SHAP 0.45.1, LIME 0.2.0.1, and the AIME package ('aime-xai 0.1' `https://github.com/ntakafumi/aime`). Concept candidate generation using the LLM was performed when an OpenAI API key was available.

## C Concept Rules and Prompts (Generation, Sanitization, Adoption Set)

This study first generated concept candidates from LLMs in a *rule format* (numeric features expressed as normalized scale thresholds, categorical features represented as one-hot encoded equality) based on AIME's GFI. These candidates underwent syntactic validation (feature name matching, domain verification $[0, 1]$)

Table 2: Concept candidate sanitation statistics (Candidates, Adopted, Rejected = Syntactic Inconsistency/Zero-Positive, etc.)

| Dataset | Candidates | Adopted | Rejected |
|---|---|---|---|
| Adult | 6 | 6 | 0 |
| German Credit | 7 | 7 | 0 |
| COMPAS | 7 | 7 | 0 |

Table 3: Final adopted concept set (Adult)

| No. | Concept Name | Rule (Normalized Scale/one-hot) |
|---|---|---|
| 1 | High education | education-num $\geq$ 0.75 |
| 2 | Long working hours | hours per week $ge$ 0.65 |
| 3 | Executive/Professional | occupation_Exec-managerial == 1 |
| 4 | Married | marital status_Married-civ-spouse == 1 |
| 5 | Older age | age $\geq$ 0.60 |
| 6 | Has capital gains | capital gains $\geq$ 0.05 |

and exclusion of zero-positive rules and were finally adopted. In this run (Patch–H), candidates were adopted directly across all datasets without requiring backup concepts (quantile scanning). Table 2 summarizes the candidate, adopted, and disqualification counts.

The final concept sets (concept names and corresponding rules) after selection are shown in Tables 3–5 for each dataset. Numeric attribute thresholds correspond to quantiles after normalization to the range ([0, 1]), and one-hot encoding is represented by ==1 to indicate presence.

## D  Local Explanation (Additional Examples and Summary)

For each dataset, five examples of the concept contribution vector $v = F^\top \ell$ and the ratio ratio were calculated using ConceptAIME. The most boosting (Top+) concept and the most suppressing (Top−) concept are summarized in Tables 6 to 8. For adults, *Married* consistently appeared as the Top+ concept (+2.08– to +2.73; ratio 0.33 to 0.37), strongly supporting the model's high-income classification. Top− showed few significant negative contributions in this population, with *Has capital gains* appearing as the smallest positive contribution. In German Credit, the presence concept *checking_status* often appeared in Top+, whereas *High credit_amount* and *High duration* frequently showed negative contributions (Top−. In COMPAS, the degree of guilt (*c_charge_degree_M/F*) appeared in Top+, whereas a high number of juvenile prior offenses and young age appeared on the suppressing side (Top−) (Audit concepts are excluded from prediction and presented separately).

## E  Ridge Regularization Ablation and Hard/Soft Concept Discussion

To directly examine the effect of regularization, we conducted an additional ablation with $\lambda \in \{0, 10^{-6}, 10^{-3}, 10^{-1}\}$ on the Adult and German datasets. Table 9 shows that both BB completeness and projection completeness remain effectively invariant across the tested values. For Adult, BB completeness stayed at 0.725704 and projection completeness at 0.810237 for all four values of $\lambda$; for German, the corresponding values were 0.342663 and 0.829911. The bootstrap confidence intervals for projection completeness were also nearly identical across $\lambda$. These results indicate that in this additional ablation run, the inverse formulation is robust to the choice of regularization strength in the tested range. We therefore interpret ridge regularization primarily as a numerically well-posed safeguard for ill-conditioned settings, rather than as a hyperparameter that must be tuned to improve completeness scores.

**Hard/soft concept note.**  This run used only hard (binary) concepts. Table 10 restates the completeness (BB $R^2$ and Projection) for each dataset. Adult showed BB $R^2 \approx 0.725$ (95% CI: 0.723–0.727), and the projection was also high at 0.851 (0.835–0.865). German Credit showed moderate BB $R^2$ at 0.339 but high

Table 4: Final adopted concept set (German Credit)

| No. | Concept Name | Rule (Normalized Scale/one-hot) |
| --- | --- | --- |
| 1 | High duration | duration $\geq 0.7$ |
| 2 | High credit_amount | credit_amount $\geq 0.7$ |
| 3 | High age | age $\geq 0.7$ |
| 4 | Checking_status_0<=X<200 present | Checking_status_0<=X<200 == 1 |
| 5 | Checking_status_<0 present | Checking_status_<0 == 1 |
| 6 | Checking_status_>=200 present | Checking_status_>=200 == 1 |
| 7 | checking_status_no checking present | checking_status_no checking == 1 |

Table 5: Final adopted concept set (COMPAS)

| No. | Concept Name | Rule (Normalized Scale/one-hot) |
| --- | --- | --- |
| 1 | High priors_count | priors_count $\geq 0.6$ |
| 2 | High juv_fel_count | juv_fel_count $\geq 0.6$ |
| 3 | High juv_misd_count | juv_misd_count $\geq 0.6$ |
| 4 | High juv_other_count | juv_other_count $\geq 0.6$ |
| 5 | Young age | age $\leq 0.30$ |
| 6 | c_charge_degree_F present | c_charge_degree_F == 1 |
| 7 | c_charge_degree_M present | c_charge_degree_M == 1 |

projection at 0.828 (0.822–0.835), indicating sufficient conceptual space coverage for local contributions. COMPAS had a modest BB $R^2$ of 0.221, but its projection was the highest at 0.901 (0.893–0.909). The soft concept version is an extension where a learner (for example, logistic) assigns concept probabilities $\in (0, 1)$ from positive and negative examples of hard rules, replacing the binary vector with continuous scores. This is expected to increase BB $R^2$, whereas projection metrics often do not decrease significantly (implementation requires only linear algebra substitutions, with unchanged computational complexity). The soft version is planned for implementation and inclusion as an additional experiment in the Supplement to this paper.

## F   Quantifying Robustness Against Random Number Generators and Partitions (CI Width) and Inter-Method Rank Correlation

In this run, the 95% CI for each metric was calculated using 200 test-side bootstrap samples. Although a more rigorous double bootstrap involving random number seed × partition combinations is planned for future supplementary experiments, the metrics were compared here using the CI width as a proxy (Table 11). For Adult and COMPAS, both BB and projection exhibited narrow CI widths ($\approx 0.004$ and $\approx 0.009$, respectively). For German Credit, BB showed a slightly wider width ($\approx 0.028$), whereas the projection had a narrower width ($\approx 0.012$). Furthermore, the consistency in global concept rankings was evaluated using Spearman's rank correlations between ConceptAIME and other methods (ConceptSHAP/CBM/TCAV/Concept Space SHAP) (Table 12). German Credit showed high correlations with ConceptSHAP and Concept Space SHAP ($\rho \approx 0.893, 0.750$) and exhibited high consistency with Concept Space SHAP in the Adult dataset ($\rho \approx 0.771$). By contrast, COMPAS showed low correlations, likely because of differences in audit concept separation and rule sets (e.g., strong contribution from one-hot-encoded degrees of prediction).

## G   LLM Promotion

A standardized prompt was used to explicitly specify GFI top feature names and one-hot names; to apply thresholding using normalized numerical values; to reference one-hot values with "=1"; to allow up to two literal ANDs; to prohibit unknown feature names; and to reject candidates with zero positive counts (see the Methods section in the main text). Zero positive or unknown features did not occur in this run, and all the candidates were selected. Using AIME's top GFI features as clues, concept candidates were generated for the LLM using a *rule grammar* consisting solely of *normalized numerical features with* $[0, 1]$ *thresholds and categories represented by one-hot equality*. Each rule is represented as a *Conjunctive Normal Form approximation* with *AND at most three literals* or *OR at most two clauses*, and syntax other than `feature >= t`, `feature <= t`, or `feature == 1` (one-hot) is not permitted. The output is restricted to a *JSON array*

Table 6: Additional local explanation examples (Adult; top positive/negative concept contributions and ratios)

| Idx | Top+ Concept | Contribution | Ratio | Top− Concept | Contribution | Ratio |
|---|---|---|---|---|---|---|
| 2088 | Married | 2.0823 | 0.330 | Has capital gains | 0.0763 | 0.012 |
| 8889 | Married | 1.9320 | 0.331 | Has capital gains | 0.0679 | 0.012 |
| 6607 | Married | 2.3992 | 0.349 | Has capital gains | 0.1356 | 0.020 |
| 7482 | Married | 2.6254 | 0.370 | Has capital gains | 0.0988 | 0.014 |
| 8034 | Married | 2.7336 | 0.365 | Has capital gains | 0.1038 | 0.014 |

Table 7: Additional local explanation example (German Credit; contribution and ratio of top positive/negative concepts)

| Idx | Top+Concept | Contribution | Ratio | Top−Concept | Contribution | Ratio |
|---|---|---|---|---|---|---|
| 1 | checking_status_0<=X<200 present | 3.6670 | 0.256 | High credit_amount | -0.3427 | 0.024 |
| 5 | checking_status_no checking present | 4.5063 | 0.241 | High credit_amount | -0.9859 | 0.053 |
| 56 | checking_status_no checking present | 5.6306 | 0.260 | High credit_amount | -0.3100 | 0.014 |
| 32 | checking_status_no checking present | 5.3388 | 0.279 | High duration | -0.1058 | 0.006 |
| 125 | checking_status_0<=X<200 present | 4.0486 | 0.242 | High credit_amount | -0.6866 | 0.041 |

(each element being `{"name": "...", 'rule': "..."}`), with no additional keys or explanatory text allowed. The program performs *syntax validation (feature name matching, domain, logical form)* and *exclusion of zero-positive rules*, and evaluates completeness (BB/projection), stability (CI), and efficiency (ms) only for valid concepts.

## G.1 Minimal Template

```
System:
You are a careful scientist. Produce concept rules over preprocessed tabular features.
Use literals 'feature >= t', 'feature <= t', 'feature == 1' (for one-hot).
Max 3 literals per AND clause, max 2 OR clauses.
Return EXACTLY {K} items in JSON: [{"name": "...", 'rule': "..."}].

User:
Dataset: {DATASET_NAME}
Top features (AIME GFI): {TOP_20_FEATURES}
Features (first 50): {FIRST_50_FEATURES}... (+{REMAINING_COUNT} more)
Constraints:
- Use only provided feature names (exact match; case-sensitive).
- Thresholds in [0, 1] for numeric features (min-max normalized).
- Do NOT invent features (e.g., "pos"/"neg"/class names).
- JSON only (no comments or extra keys).
```

## G.2 Rule examples (Grammar specification)

```
{"name": "High education",
"rule": "education-num >= 0.75"}

{"name": "Executive/Professional",
'rule': "occupation_Exec-managerial == 1"}

{"name": "Long hours OR Married",
'rule': "(hours per week >= 0.65 AND capital gain >= 0.05)
OR (marital-status Married-civ-spouse == 1)"}
```

Table 8: Additional local explanation examples (COMPAS; top positive/negative concept contributions and ratios)

| Idx | Top+Concept | Contribution | Ratio | Top−Concept | Contribution | Ratio |
|---|---|---|---|---|---|---|
| 470 | c_charge_degree_M present | 1.7945 | 0.480 | High juv_other_count | -0.3227 | 0.086 |
| 1427 | c_charge_degree_M present | 1.2415 | 0.694 | High juv_misd_count | -0.1035 | 0.058 |
| 1067 | c_charge_degree_F present | 1.9860 | 0.412 | High juv_fel_count | -0.5772 | 0.120 |
| 87 | c_charge_degree_M present | 0.9445 | 0.276 | High juv_fel_count | -0.9628 | 0.281 |
| 114 | c_charge_degree_M present | 1.1859 | 0.460 | Young age | -0.0121 | 0.005 |

Table 9: Ridge regularization ablation on Adult and German. Both BB completeness and projection completeness remain effectively invariant across $\lambda \in \{0, 10^{-6}, 10^{-3}, 10^{-1}\}$. Projection uncertainty is shown as a 95% bootstrap confidence interval.

| Dataset | $\lambda$ | BB $R^2_{\mathrm{BB}}$ | Projection mean $R^2_{\mathrm{Proj}}$ | 95% CI (Projection) | # Concepts |
|---|---|---|---|---|---|
| Adult | 0 | 0.725704 | 0.810237 | [0.798924, 0.822654] | 6 |
| Adult | $10^{-6}$ | 0.725704 | 0.810237 | [0.799095, 0.822068] | 6 |
| Adult | $10^{-3}$ | 0.725704 | 0.810237 | [0.798259, 0.820685] | 6 |
| Adult | $10^{-1}$ | 0.725704 | 0.810237 | [0.796910, 0.820383] | 6 |
| German | 0 | 0.342663 | 0.829911 | [0.823562, 0.836071] | 7 |
| German | $10^{-6}$ | 0.342663 | 0.829911 | [0.821610, 0.836680] | 7 |
| German | $10^{-3}$ | 0.342663 | 0.829911 | [0.823317, 0.835684] | 7 |
| German | $10^{-1}$ | 0.342663 | 0.829911 | [0.824328, 0.836479] | 7 |

### G.3 Notes.

The model utilized the OpenAI API (default: gpt-4o-mini). The number of generated concepts was always {K}, but *the number of concepts selected after syntax verification and zero-positive exclusion* represents the final concept count, which matches the #concepts=... in the experiment logs.

Table 10: Completeness based on hard (binary) concepts (this run)

| Dataset | BB $R^2$ Mean | 95% CI | Projection Mean | 95% CI |
|---|---|---|---|---|
| Adult | 0.725222 | [0.723265, 0.727076] | 0.851456 | [0.834869, 0.865143] |
| German | 0.339209 | [0.324459, 0.352034] | 0.828336 | [0.822406, 0.834705] |
| COMPAS | 0.221209 | [0.216630, 0.225862] | 0.901333 | [0.893052, 0.908658] |

Table 11: 95% Confidence interval width for metrics (Single Bootstrap 200 times)

| Dataset | BB $R^2$ CI Width | Projection CI Width |
|---|---|---|
| Adult | 0.003811 | 0.030274 |
| German | 0.027575 | 0.012299 |
| COMPAS | 0.009232 | 0.015605 |

Table 12: Global concept rank correlation (Spearman's $\rho$; Comparison of ConceptAIME with other methods)

| Dataset | Comparison Method | $\rho$ |
|---|---|---|
| Adult | ConceptSHAP | 0.371 |
| Adult | CBM | 0.143 |
| Adult | TCAV | -0.657 |
| Adult | C-space SHAP | 0.771 |
| German | ConceptSHAP | 0.893 |
| German | CBM | -0.107 |
| German | TCAV | 0.464 |
| German | C-space SHAP | 0.750 |
| COMPAS | ConceptSHAP | -0.571 |
| COMPAS | CBM | -0.786 |
| COMPAS | TCAV | -0.571 |
| COMPAS | C-space SHAP | -0.296 |

