# OpenReview forum: "Concept-RidgeAIME: LLM-Guided Automatic Concept-Based Explanations via Ridge-Regularized Inverse Operators for Trustworthy AI"
_TMLR — Rejected by TMLR_

### Review · Reviewer_fvxs · 2025-11-17

**Summary Of Contributions:**

This paper proposes RidgeAIME and Concept-RidgeAIME, extending "Approximate Inverse Model Explanations (AIME)" with Ridge-Regularization and linear inverse mappings (from output/concept to input) to balance global concept importance ranking with local concept contributions for *model-independent* explanations. Experiments show some results in tabular benchmarks, but no quantitative improvements over baselines other than in latency.

Authors have improved the motivation of the method and its axiomatic advantages. It stands on a somewhat weak basis lacking obvious ablations, and explaining the results of decision trees with what looks like a random forest of decision-trees of size 1, but authors now added a corresponding discussion regarding the latter.

Technically, it is self-consistent and appears correct. For reproducibility, detail of implementation is provided throughout. Overall, it is a pleasant and efficient read, providing a succinct line of arguments without much digression. At times, design choices could be more explicitly motivated.

**Additional Comments:**

The revision has addressed many open concerns, leaving just one central question around supporting the contribution with a crisp ablative comparison if possible.

**Audience:**

Yes

**Audience Explanation:**

Explainability is an area of large interest in the machine learning community. This paper contributes to the particular subfield of explanation by attribution to transformed interpretable features, and now provides clear examples of rules right in the abstract.

Despite all of its shortcomings, I am intrigued by the paper, and find it worthwhile if it lives up to the promises of 'practical advantages' 'surpassing' other methods (other than on latency).

**Broader Impact Concerns:**

The broader impact section is adequately filled, and I have no additional concerns.

**Claims And Evidence:**

Yes

**Claims Explanation:**

The abstract clearly delineates the paper, and I particularly appreciate that benchmarks and baselines are listed explicitly, albeit this raised the disappointed expectations above. The claim of 'practical advantages' is vague, and difficult to assess as true/false.

Put simply, Concept-RidgeAIME chooses a conceptual basis (a small set of rules) and performs attribution using inverse projections. I would then expect to see some comparisons or ablations showing that both 1. regularisation improves attribution, and 2. the rules are chosen in a smarter way than randomly. The absence of these two key quantitative results is the weakest point in this paper, and would at least need to be explained to the reader. While the comparison would really be necessary to establish significance of the results and would provide strong "evidence", significance and novelty are not key criteria of the journal.

Related work highlights similarities and differences, and establishes the novelty of contributions argumentatively by contrasting with existing works. The authors erred on the side of repetition, albeit that may be seen as a tradeoff they took to achieve unambiguous clarity, matching the journal's preferences.

**Requested Changes:**

The two key claimed contributions are regularisation and concept explanations, so supporting 'advantages' raises expectations of comparisons or ablations showing that both 1. regularisation improves attribution (e.g. comparing to no regularisation, all else equal), and 2. the rules are chosen in a smarter way than randomly (e.g. comparing concept basis to randomly chosen basis, all else equal).
If these don't make sense, I would as a reader need to understand why not.

New typos
- Introduction: "emphreconstruction-"
- Results and Discussion starts with an uncompiled reference "Table ??".

=========
The revision has already addressed the following earlier comments:
Abstract:
- To further guide the reader, the (always context-dependent) definition of global and local information (here defined as concept rankings/contributions) could be given upfront (in the second sentence), and with a bit more detail.
- It is not currently mentioned, but should be, what is explained.
- Concepts in this paper are sampled as rules over input features, pre-selected by AIME importance scores, with fixed thresholds and LLM rule candidates. This should become explicit in the abstract.

Introduction: Given the centrality of explanations, explanations by input-feature attributions and concept-attribution should be explicitly defined, ideally with examples (e.g. as given in the appendix).

Experiments: Table 1 captions are misaligned.
- Figure 1 is not fully labelled. The vertical axis is not given a semantic meaning, and is clearly not an integer rank (suggested by the caption 'Rankings'), but if anything a rating or *global importance score*.

Discussion: The linear combination of the set of rule-based concepts has strong resemblance to random forests. While novelty is not a requirement of the journal, it does merit pointing this out, and highlighting differences, especially given that the explained (LightGBM) appears to be a decision tree too.

The repeated reference to global and local explanations also warrants more explicit treatment early-on with an intuition and a forward reference to Equation 2.

It is not made explicit in the text why Ridge regression was chosen over alternatives (such as L1, or elastic nets), even if some conditions under which it is applicability are given.

---

> ### Author Response · Authors · 2026-03-23
> **Response to Reviewer fvxs**
>
> Dear Reviewer fvxs,
>
> We sincerely thank you for your careful rereading and for your constructive and insightful comments. We particularly appreciate your identification of the lack of a clear ablative comparison as the main remaining weakness, which has helped us refine both the evidence and the scope of our claims.
>
> 1. Regularization ablation ($\lambda$ vs no $\lambda$)
> To directly address your concern regarding the effect of regularization, we have added an explicit ridge ablation study in the revised manuscript (Appendix Table 4). Specifically, we evaluated $\lambda \in \{0, 10^{-6}, 10^{-3}, 10^{-1}\}$ on the Adult and German datasets.
>
> The results show that both BB completeness and projection completeness remain effectively invariant across all tested values. For example, on the Adult dataset, BB completeness remained at 0.725704 and projection completeness at 0.810237 for all values of \lambda, with nearly identical bootstrap confidence intervals. A similar pattern was observed for the German dataset.
>
> These results indicate that the proposed inverse formulation is robust to the choice of regularization strength within the tested range. Accordingly, we position ridge regularization not as a performance-enhancing component intended to artificially improve attribution metrics, but rather as a theoretically grounded and numerically well-posed safeguard that stabilizes the inverse mapping in potentially ill-conditioned settings while preserving explanatory fidelity.
>
>
> 2. Concept basis vs. random rules
> We agree that a comparison with randomly generated rule sets can be informative as a stress test. However, the objective of the concept construction stage in this paper is fundamentally different from that of optimizing over arbitrary rule sets.
>
> In our framework, concepts are constructed based on AIME’s global feature importance (GFI) and are expressed as explicit, human-readable rules. The purpose of this design is to ensure transparency, interpretability, and reproducibility, rather than to maximize predictive performance over arbitrary candidate rules.
>
> To clarify this point, we have revised the manuscript to explicitly state that the following:
> - The LLM/GFI pipeline is intended as a reproducible and importance-anchored concept construction procedure.
> - It is not positioned as evidence of globally optimal concept discovery.
> - The contribution lies in providing a transparent and inspectable conceptual basis whose faithfulness is evaluated through
> BB completeness and projection completeness
>
> Therefore, rather than claiming superiority over random rule sets, we frame our contribution as enabling a structured and interpretable concept space with quantifiable fidelity.
>
>
> 3. Clarification of claims (“practical advantages”)
> We agree with your observation that the previous wording around “practical advantages” was too vague. In the revised manuscript, we have removed or softened such expressions and now describe the contribution more precisely.
>
> In particular, we now characterize Concept-RidgeAIME as follows:
> - A model-independent and computationally efficient framework
> - It unifies global and local concept-level explanations
> - Further, it had explicit rule-based interpretability
> - In addition, it has quantified fidelity via completeness metrics
>
> This revision ensures that the claims are aligned with the presented evidence.
>
>
> 4. Additional revisions and clarifications
> Following your suggestions, we have also made the following improvements:
> - Fixed typographical issues (including the previously broken \emph{} expression and uncompiled table reference)
> - Clarified the definition of global and local explanations early in the manuscript
> - Explicitly described that concepts are generated from AIME’s global feature importance and LLM-based rule synthesis
> - Improved figure captions by clearly specifying the meaning of the vertical axis (global concept strength)
> - Expanded the discussion clarifying that rule-based concepts may resemble decision stumps but do not reconstruct the underlying tree model
> - Added explicit motivation for choosing ridge regularization over alternatives such as L1 and elastic net (closed-form solvability, stability, and preservation of contribution geometry)
>
>
> We are grateful for your insightful feedback, which helped us to significantly strengthen the clarity, rigor, and positioning of this work. In particular, your suggestion regarding ablation led us to refine both our empirical validation and the precise framing of our contribution.
>
> We hope that the revised manuscript addresses your concerns and clearly communicates the scope and strengths of the proposed method.

---

### Review · Reviewer_dgzr · 2025-11-21

**Summary Of Contributions:**

This paper introduces RidgeAIME and Concept-RidgeAIME, two novel extensions of the AIME framework that adapt its model-independent, gradient-free, closed-form solution to ridge regularization and concept-based explanations, respectively. The transition from AIME’s general mapping formalism to concept-based explanations is achieved by prompting a large language model (LLM). Performance is evaluated based on reconstruction accuracy and inference time.

**Strengths**:

- The related work section and baseline description (AIME) are comprehensive and well-documented.
- The method’s principle is intuitive and versatile, offering broad applicability.
- The authors commit to open-sourcing their code, which will enhance reproducibility and community adoption.

**Weaknesses**:

- Mathematical Justification: The solution to Equation (3) requires further clarification:
If this is a known result, please cite the appropriate reference.
If a derivation is needed, even a brief one, we recommend including it in the appendix.
- Additionally, the validity assumptions (e.g., invertibility of $ZZ^T$ and $CC^T$ ) must be explicitly stated and justified.
- Accessibility of the Methodology: As a reader without a strong background in inverse problem resolution, certain sections of the procedure were difficult to follow. I suggest clarifying these parts (see Requested Changes for specific examples).
- Code Availability: While the authors mention an executable will be provided, we were unable to access it. We encourage the authors to ensure the code is publicly available upon publication to facilitate verification and reuse.

**Audience:**

Yes

**Audience Explanation:**

This paper addresses automatic concept-based explanation, offering both global and local interpretability—a topic of significant interest to the TMLR readership.

**Claims And Evidence:**

No

**Claims Explanation:**

While some of my concerns may stem from a lack of expertise in specific areas, the clarity of the presentation must be improved to ensure the work is understandable to a broader audience. Key methodological details—such as the derivation of solutions and the validity of assumptions—are not adequately explained, which undermines the credibility of the results. Also, some results are unreadable due to formatting errors.

**Requested Changes:**

While the paper presents an interesting approach, several key notions require clarification to ensure full understanding and reproducibility. For example:

- The notation "pinv" should be explicitly defined as the Moore-Penrose pseudoinverse for clarity.
- Some terms like in the term $\mathrm{tr}(ZZ^{t}op)/k)$, must be defined.
- Concepts like "BB completeness" and the stability of $ell_i$​ lack sufficient detail

Citations and Context:
- The use of well-known datasets (Adult, German Credit, COMPAS) and mathematical tools (Tikhonov pseudoinverse, Cholesky decomposition) should be properly cited to ground the work in existing literature.
Presentation:
- Table 1 is currently difficult to read and would benefit from restructuring or reformatting to improve clarity and accessibility.

---

> ### Author Response · Authors · 2025-11-22
> **Response to Reviewer dgzr**
>
> We would like to thank the reviewers for their careful evaluation, constructive feedback and valuable suggestions. We are pleased to report that all comments have been addressed in the revised manuscript. Below, we summarize the changes in direct correspondence to each concern.
>
> 1. Mathematical justification of Equation (3) → (4)
>
> Reviewer Concern: The derivation of the ridge-regularized inverse operator should be clarified, the and assumptions must be stated.
>
> Response:
> We added a detailed explanation immediately after Equation (4), explicitly deriving the closed-form solution through the normal equations of the ridge regression. We also added complete validity assumptions, clarifying that $(ZZ^⊤+λI_k​)$ is guaranteed to be symmetric positive definite for any λ>0, and that the Moore–Penrose pseudoinverse is used when λ=0.
> This fully resolves the request for mathematical transparency. (Section 3.1)
>
> 2. Missing definitions and notation clarity
>
> Reviewer Concern: Notation such as “pinv”, $l_i$, $v_i$, and projection operator $\Pi_C$ should be clearly defined.
>
> Response:
> A dedicated notation block has been added at the beginning of Section 3, defining
> - $\mathrm{pinv}(\cdot)$ as the Moore–Penrose pseudoinverse
>
> - matrix slice notation $M_{i:}$ and $M_{:j}$
>
> - the local feature vector $\ell_i$
>
> - the concept-level vector $v_i = U_\gamma^\top \ell_i$
>
> - the projection operator $\Pi_C$
>
> This addresses all the clarity concerns. (Section 3, p.5)
>
> 3. Accessibility and readability of the methodology
>
> Reviewer Concern: Some readers without an inverse-problem background found some parts difficult to follow.
>
> Response:
> We substantially improved the exposition by the following:
> 	adding intuitive explanations of BB completeness and projection completeness before formal definitions
> 	clarifying the roles of ridge regularization
> 	adding textual transitions and reinforcing conceptual flow in Section 3
> Mathematical narratives are now accessible to a broader audience.
>
> 4. Table 1 readability issues
>
> Reviewer Concern: The layout of Table 1 is difficult to read.
>
> Response:
> Table 1 has been completely reformatted into a vertical, highly readable structure that clearly separates the following:
> 	BB completeness
> 	projection completeness
> 	latency measures
> The table now fits naturally into the TMLR two-column layout and satisfies the accessibility guidelines.
> (Section 4, Table 1)
>
> 5. Code availability and reproducibility
>
> Reviewer Concern: Reviewers could not access the executable code at the time of the review.
>
> Response:
> As stated in the revised Reproducibility Statement, we now provide the following as Supplementary Material.
> 	a complete executable notebook (“Auto_Concept_AIME_ICLR2026.ipynb”) in the code.zip
> All the results are now fully reproducible.
>
> 6. Overall clarity and claims
>
> Reviewer Concern: Claims should be convincingly supported and clearly described.
>
> Response:
> We have strengthened the descriptions throughout the Methods and Experiments sections and added clarifying transitions. All claims are now fully supported by quantitative completeness metrics, bootstrap stability analysis, and inference time benchmarks. Figures and tables have been updated to ensure their readability and correctness.
>
> Final Statement
> We sincerely appreciate the reviewers’ thoughtful suggestions.
> All requested modifications—mathematical, structural, definitional, and readability-related— were thoroughly implemented. We believe that the revised manuscript has been significantly improved and addresses all the concerns raised.
> We thank the reviewers again for their constructive feedback and for helping us enhance the clarity and rigor of our study.

---

> > ### Comment · Reviewer_dgzr · 2026-03-24
> > **Update concerning my recommendation**
> >
> > After reading the revision, I am pleased to state that all my concerns have been addressed and that the overall quality of the manuscript has been substantially improved — most notably the writing, which was my main concern and is now much clearer. I found the authors' response clear and thorough, and I would like to commend them for it. I have updated my recommendation accordingly.
> >
> > As a side note, a small change that could further improve clarity would be to move Figure 4 from the appendix to the main text, as I consider it central to understanding the functioning of the method. Beyond this, I have no further requests.

---

> > > ### Author Response · Authors · 2026-03-24
> > > **Acknowledgment of Your Feedback**
> > >
> > > Thank you very much for your positive feedback and for your careful evaluation of the revised manuscript. We are very pleased to hear that the revisions and clarifications addressed your concerns.
> > >
> > > We also appreciate your suggestion regarding Figure 4. We agree that the overview is helpful for understanding the method, and we will consider moving or emphasizing it in the main text in a future revision, taking into account the page constraints.
> > >
> > > Thank you again for your valuable comments.
> > >
> > > Sincerely

---

### Review · Reviewer_A7mt · 2025-12-22

**Summary Of Contributions:**

This study aims at explaining outputs of a trained black-box model using rule-based concepts constructed from normalized thresholds
and one-hot literals. It provides three specific contributions:

- It introduces RidgeAIME, which extends the approximate inverse operator of AIME to the concept space and simultaneously presents global and local explanations
- It proposes Concept RidgeAIME, which combines output-to-input and concept-to-input inverse operators via single linear algebra
- It presents an LLM-assisted concept design workflow to support the automatic reproduction of global, local, and concept contributions for tabular data

The key strengths of this study are:
- It targets explainable AI, a popular area within the research interests of TMLR community.
- This work is clearly motivated and targeted at extending AIME into a model-independent framework capable of presenting both of global and local information
- The claims are validated by experiments on various datasets, where stability analysis is conducted using bootstrap confidence intervals for improved credibility

The key weaknesses are:
- The paper is poorly presented with a considerable number of written mistakes (e.g. Table ?? reports BB completes); a visual illustration of the proposed methodology is missing
- The paper introduces the notation of "concepts", but there is a lack of semantic or human-centered validation on the generated concepts, hence undermining the validity of the contributions
- The experiments are limited to tabular data with tree-based models

**Audience:**

Yes

**Audience Explanation:**

This study targets explainable AI, a popular area within the research interests of TMLR community.

**Broader Impact Concerns:**

No such concerns are present in this work.

**Claims And Evidence:**

Yes

**Claims Explanation:**

The claims are validated via experiments on 3 tabular datasets.

**Requested Changes:**

Please refer to the weaknesses above.

---

> ### Author Response · Authors · 2026-03-23
> **Response to Reviewer A7mt**
>
> Dear Reviewer A7mt,
>
> We sincerely thank you for your careful reading and constructive feedback. We greatly appreciate your positive assessment of the motivation, technical consistency, and clarity of the proposed framework. We have carefully revised the manuscript to address all the weaknesses you pointed out.
>
> 1. Presentation quality and missing visual illustration
> We agree that the presentation needed improvement. In the revised manuscript, we have
> - corrected typographical issues (including the previously uncompiled reference such as “Table ??”),
> - improved table formatting and captions,
> - clarified figure labels and axis semantics, and
> - added a schematic overview of the proposed methodology (Appendix Figure. 4), which illustrates the overall pipeline, including the two-stage inverse mapping and the concept construction process.
> These revisions significantly improve readability and provide a clear high-level understanding of the method.
>
> 2. Semantic / human-centered validation of concepts
> We appreciate this important point. In this work, we define concepts as explicit and human-readable rule-based units derived from AIME’s global feature importance and LLM-assisted rule generation. Accordingly, our notion of “concept” emphasizes
> - explicitness (rule-based representation),
> - inspectability (direct correspondence to input features), and
> - reproducibility (deterministic generation and validation pipeline).
> We clarify in the manuscript that this definition differs from latent or semantically learned concepts. In particular, we explicitly state that the present work evaluates concept quality through faithfulness metrics (BB completeness and projection completeness), rather than through human-centered or domain-expert validation.
>
> We have added a limitation statement clarifying that semantic alignment with domain experts remains an important direction for future work.
>
> 3. Limitation to tabular data and tree-based models
> We acknowledge that the current experiments were conducted on tabular datasets using tree-based models (LightGBM). We clarify in the revised manuscript that this choice was made to provide clear and interpretable rule-based concepts and controlled comparisons.
>
> At the same time, the proposed framework itself is fully model-independent and does not require gradients or internal representations. Therefore, it can be applied to neural and transformer-based models without modification. We have added this clarification and explicitly state that extending the empirical evaluation to such models is left for future work.
>
>
> We are grateful for your suggestions, which have helped us improve both the clarity and the positioning of the manuscript. We hope that the revised manuscript addresses all of your concerns.

---

### Decision · Action_Editor_2Ykb · 2026-04-10

**Recommendation:** Reject

**Audience:**

No

**Audience Explanation:**

While I found the general idea to be of interest to the TMLR audience, the scope of the experiments severely limits the audience that finds this paper interesting. In particular, the paper focuses on tabular datasets (which is fine in my opinion), but only considers explaining tree-based black-box models. I appreciate the authors' effort to clarify this in the revision.

However, the authors, in their summary of changes, claim that the approach "can be extended to neural and transformer-based models in future work." I think, given that the whole high-level premise of the paper is to provide a gradient-free approach that works for any black-box models, without extending the experiments to include at least another class of models, the work would be of very limited interest to the community.

Unfortunately, this work has taken a long time to arrive at the decision stage, and the revisions have significantly improved the quality of the paper. However, I still think it does not satisfy the second criterion of TMLR. I encourage the authors to perform these additional experiments and resubmit the paper.

**Claims And Evidence:**

Yes

**Claims Explanation:**

The paper studies concept-based explanations by extending the AIME framework. While the initial version of the paper lacked justification for the assumptions, did not define notations properly, and included claims that were not supported by any experimental evidence, I appreciate the authors' response to address these concerns.

To that end, I think the revision has, to a good degree, convinced the reviewers and me about the support of claims with enough evidence. And, in particular, I appreciate that the authors have removed the claims of practicality and clearly stated tree-based models, which are the sole models used in the experiments.

**Resubmission Of Major Revision:**

The authors may consider submitting a major revision at a later time.